# Effects of Phase Structure Regulation on Properties of Hydroxyl-Terminated Polyphenylpropylsiloxane-Modified Epoxy Resin

**DOI:** 10.3390/polym17152099

**Published:** 2025-07-30

**Authors:** Yundong Ji, Jun Pan, Chengxin Xu, Dongfeng Cao

**Affiliations:** 1School of Materials Science and Engineering, Wuhan University of Technology, Wuhan 430070, China; jiyundong@whut.edu.cn (Y.J.); 18326305543@163.com (J.P.); xu226316@163.com (C.X.); 2State Key Laboratory of Advanced Technology for Materials Synthesis and Processing, Wuhan University of Technology, Wuhan 430070, China; 3Foshan Xianhu Laboratory of the Advanced Energy Science and Technology Guangdong Laboratory, Foshan 528000, China; 4Institute of Advanced Materials and Manufacturing Technology, Wuhan University of Technology, Wuhan 430070, China

**Keywords:** polysiloxane, epoxy resin, characterization, compatibilizer, phase structure, toughening and strengthening

## Abstract

4,4’-Methylenebis(N,N-diglycidylaniline) (AG80), as a high-performance thermosetting material, holds significant application value due to the enhancement of its strength, toughness, and thermal stability. However, conventional toughening methods often lead to a decrease in material strength, limiting their application. Modification of AG80 epoxy resin was performed using hydroxy-terminated polyphenylpropylsiloxane (Z-6018) and a self-synthesized epoxy compatibilizer (P/E30) to regulate the phase structure of the modified resin, achieving a synergistic enhancement in both strength and toughness. The modified resin was characterized by Fourier transform infrared analysis (FTIR), proton nuclear magnetic resonance (^1^H NMR) spectroscopy, silicon-29 nuclear magnetic resonance (^29^Si NMR) spectroscopy, and epoxy value titration. It was found that the phase structure of the modified resin significantly affects mechanical properties. Thus, P/E30 was introduced to regulate the phase structure, achieving enhanced toughness and strength. At 20 wt.% P/E30 addition, the tensile strength, impact strength, and fracture toughness increased by 50.89%, 454.79%, and 152.43%, respectively, compared to AG80. Scanning electron microscopy (SEM) and atomic force microscopy (AFM) analyses indicate that P/E30 regulates the silicon-rich spherical phase and interfacial compatibility, establishing a bicontinuous structure within the spherical phase, which is crucial for excellent mechanical properties. Additionally, the introduction of Z-6018 enhances the thermal stability of the resin.

## 1. Introduction

As a significant thermosetting resin, 4,4’-methylenebis(N,N-diglycidylaniline) (AG80) exhibits extensive applications in automotive, aerospace, and electronics industries, attributable to its exceptional chemical stability, thermal resistance, adhesive properties, and electrical insulation performance [1]. However, the high crosslinking density of cured AG80 resin exhibits inherent brittleness, leading to fracture under impact loading. Additionally, the molecular structure of AG80 contains C-N bonds with low bond energy (305 kJ/mol), significantly lower than C−O bonds (346 kJ/mol) and C−C bonds (357 kJ/mol), resulting in reduced char yield. These defects limit its broader applications in industrial fields. Current research on toughening epoxy resins primarily employs thermoplastic polymers, rubber-based modifiers, and inorganic nanofillers [2]. However, these methods often compromise tensile strength and thermal stability to varying degrees. Consequently, achieving synergistic enhancement in both mechanical properties and thermal performance remains a critical challenge.

Introducing polysiloxanes is recognized as an effective strategy to enhance both the toughness and thermal stability of epoxy resins [3]. Linear polysiloxanes ([R_2_SiO]ₙ, R = −CH_3_, −C_6_H_5_) are polymers with a Si−O−Si backbone and organic side groups. Polysiloxanes with reactive terminal groups (e.g., hydroxyl, amino, alkoxy groups) can react with epoxy or hydroxyl groups in epoxy resins, thereby functioning as effective modifiers. Research indicates that the inclusion of polysiloxane as a modifier in epoxy resin systems can effectively enhance material toughness but may lead to a reduction in tensile strength. Zhang and Selvaganapathi [4,5] modified AG80 epoxy resin using amino-bearing phenyl silicone resin (APSR) and hydroxy-terminated polydimethylsiloxane (HTPDMS), respectively. With 25 wt.% APSR, the modified resin exhibited 45%, 193%, and 34.6% increases in fracture toughness, impact strength, and char yield, respectively. However, 15 wt.% HTPDMS enhanced impact strength by 23.7% but reduced tensile strength by 26.1%. In terms of performance results, APSR exhibits superiority over HTPDMS, primarily attributed to: (1) the strong amino reactivity, enhancing the interface binding between the dispersed phase and the matrix, and (2) the presence of the benzene ring, which enhances polarity. The enhancement in the mechanical properties of the modified resin originates from the evolution of the phase structure during curing, while the increase in char yield is ascribed to the formation of a silicon-rich carbon layer by organic silicon, thereby retarding the resin’s pyrolysis. HTPDMS-modified epoxy resin failed to achieve a synergistic improvement in both toughness and strength. Regulating the phase structure and enhancing interfacial interactions may concurrently enhance toughness, strength, and thermal stability.

Cai et al. [6] utilized modified thermoplastic polymer (MTP30) to toughen AG80 epoxy resin. The impact strength of the modified resin increased by 109% compared to the unmodified resin when the MTP30 content was 15 phr, with a slight reduction in Tg, but was maintained at 221 °C. SEM observation revealed that MTP30 was uniformly dispersed in the resin matrix, inducing crazes and shear bands, which effectively absorbed impact energy through a multiphase structure, thereby achieving toughening. However, the study lacked tensile strength data.

Polysiloxane modification of E51 epoxy resin similarly leads to enhanced toughness but reduced tensile strength. Ren and Heng et al. [7,8] modified epoxy resin (E51) with polymethylphenylsiloxane (DC-3074) and methylphenylsiloxane intermediate (PMPS), respectively, fabricating two polysiloxane-modified epoxy systems (E3074/PSG). Both modified systems exhibited enhanced toughness and thermal stability. However, the tensile strength of E3074 and PSG decreased by 83.3% and 66.7%, respectively. SEM observations revealed that the size of silicon-rich spherical phases significantly influenced the mechanical properties.

According to previous studies, enhancing the toughness and thermal stability of epoxy resins can be achieved by utilizing various polysiloxane-modified resins. However, the improvement in toughness often comes at the expense of reduced tensile strength, a phenomenon known as “toughness without strength”. This is attributed to (1) the presence of a soft phase lowering the crosslink density, leading to decreased network rigidity, and (2) poor compatibility between toughening agents and the matrix, resulting in weak interfacial interactions. Consequently, under external forces, cracks propagate along the weak interfaces, causing material fracture [9]. This impedes the simultaneous achievement of strength and toughness in modified resins. Previous studies by our team revealed that hydroxyl-terminated polyphenylpropylsiloxane (PPPS) simultaneously enhanced the toughness and strength of bisphenol A epoxy resin (E51). Ji and Wang attributed this synergistic improvement to the formation of co-continuous phases during resin curing [10,11,12]. Currently, there are three viewpoints regarding the reasons for the strength improvement of toughened epoxy resin: (1) strong interfacial bonding facilitates stress transfer from the matrix to the toughener, thereby increasing tensile strength [13]; (2) hydrogen bonding reinforces intermolecular interactions [14]; (3) toughener incorporation alleviates internal residual stresses [15]. Based on the above viewpoint, by introducing hydroxyl-terminated polyphenylpropylsiloxane into AG80 epoxy resin to alleviate internal stress, and utilizing the epoxy toughening agent P/E30 synthesized by our team to regulate the phase structure, it is anticipated that an enhancement in toughness, tensile strength, and thermal stability could be achieved.

In this study, AG80 epoxy resin was modified with Z-6018. The molecular structure was characterized by FT-IR, ^1^H NMR, and ^29^Si NMR. To enhance AG/P resin compatibility, the epoxy compatibilizer P/E30 (synthesized by our team) was introduced, with phase morphology analyzed using SEM and AFM. Thermal stability and dynamic mechanical properties were investigated by thermogravimetric analysis (TGA) and dynamic mechanical analysis (DMA). The novelty of this work lies in elucidating the relationship between phase structure and mechanical properties and proposing a methodology to enhance mechanical properties by regulating the phase structure of modified resins via epoxy compatibilizer introduction. This approach can be extended to other epoxy/polysiloxane systems, providing a viable strategy for developing resin matrices with balanced high strength, toughness, and thermal stability.

## 2. Materials and Methods

### 2.1. Experimental Materials

4,4’-methylenebis(N,N-diglycidylaniline) (AG80, industrial grade, with an epoxy value of 0.85 mol/100 g, purchased from Chuzhou Huisheng Electronic Materials Co., Ltd., Anhui, China); hydroxyl-terminated polyphenylpropylsiloxane (Z-6018, industrial grade, Mn = 2500, purchased from Shanghai Dow Corning Co., Ltd., Shanghai, China); dibutyltin dilaurate (DBDTL, analytical grade, purchased from Sinopharm Chemical Reagent Co., Ltd., Shanghai, China); epoxy compatibilizer (P/E30, synthesized by our team); 4-Methylphthalic anhydride (4-MHHPA, analytical grade, purchased from Sinopharm Chemical Reagent Co., Ltd., Shanghai, China); 2,4,6-Tris(dimethylaminomethyl)phenol (DMP-30, analytical grade, purchased from Sinopharm Chemical Reagent Co., Ltd., Shanghai, China); acetone (analytical grade, purchased from Sinopharm Chemical Reagent Co., Ltd., Shanghai, China).

### 2.2. Synthesis of AG/P Resin

AG80 epoxy resin and Z-6018 in varying weight ratios were dissolved in acetone and added to a 500 mL three-necked flask. A thermometer and a stirring rod were inserted, and one neck was sealed with a glass stopper to prevent solvent evaporation. After stirring at 200 r/min for 20 min, DBDTL catalyst was added, and the mixture was allowed to react at 25 °C for 24 h. Upon completion of the reaction, the product was transferred to a rotary evaporator to remove the solvent, yielding AG/P resin. Based on the different weight percentages of Z-6018 (10 wt.%, 20 wt.%, and 30 wt.%), the samples were named AG/P10, AG/P20, and AG/P30, respectively. Figure 1 illustrates the synthesis process of AG/P.

### 2.3. Preparation of AG/P30-S Resin

Self-synthesized P/E30 epoxy compatibilizer and AG/P30 resin were charged into a 250 mL three-necked flask at varying mass ratios, equipped with a thermometer and mechanical stirrer. The remaining neck was sealed with a glass stopper. The mixture was homogenized at 60 °C and 200 rpm for 20 min to yield AG/P30-S resin. Based on the different weight percentages of P/E30 (10 wt.%, 20 wt.%, and 30 wt.%), the samples were named AG/P30-S10, AG/P30-S20, and AG/P30-S30, respectively.

### 2.4. Curing of Resin

Based on the test data of epoxy value, the curing agent 4-MHHPA and accelerator DMP-30 were added. The resin, curing agent, and accelerator were homogenized by mechanical stirring; the mixture was then poured into the mold and transferred to an oven for curing. The curing profile was 100 °C/3 h + 120 °C/3 h + 150 °C/2 h + 180 °C/2 h + 200 °C/2 h. The formulation of the cured resin is listed in Table 1.

### 2.5. Characterization Methods

#### 2.5.1. Fourier Transform Infrared

Using a Fourier transform infrared spectrometer (6700, Thermo Fisher Scientific, Waltham, MA, USA). The uncured resin was uniformly coated onto a KBr window, avoiding bubble formation to ensure sufficient light transmission. Scanning was performed at room temperature, with a wavenumber range of 4000−400 cm^−1^.

#### 2.5.2. Nuclear Magnetic Resonances

NMR spectroscopy was performed using a Bruker AVANCE III 400 MHz spectrometer (Bruker, Germany). Deuterated chloroform (CDCl_3_) served as the solvent for both ^1^H NMR and ^29^Si NMR measurements.

#### 2.5.3. Epoxy Value Titration

Epoxy value titration was performed according to Method II (hydrochloric acid–acetone–phenolphthalein) in GB/T 1677-2023 [16]. Precisely measured 0.5 g–1.0 g of the sample (accurately weighed to 0.0001 g) was placed into a 250 mL glass-stoppered conical flask. Then, 20.00 mL of hydrochloric acid–acetone solution (1:40 *v*/*v*) was added, and the flask was sealed, homogenized by shaking, and stored in darkness for 30 min. After adding 3 drops of phenolphthalein indicator (10 g/L), the mixture was titrated with 0.15 mol/L NaOH standard solution until a faint pink endpoint persisted for 30 s. The consumed volume of NaOH solution was recorded. A blank test was conducted simultaneously following identical procedures.

The epoxy value is calculated using the following formula:$$W=\frac{\left[V_{2}-\left(V_{3}-\frac{V_{4}}{m_{2}}\times m_{3}\right)\right]\times c\times 15.999}{1000\times m_{3}}\times 100\%$$


w—The epoxy value mass fraction.

V_2_—The volume of sodium hydroxide standard titration solution consumed in the blank test, in milliliters (mL).

V_3_—The volume of sodium hydroxide standard titration solution consumed in the sample test, in milliliters (mL).

V_4_—The volume of sodium hydroxide standard titration solution consumed in acid value determination, in milliliters (mL).

m_2_—The mass of the sample in acid value determination, in grams (g).

m_3_—The mass of the test sample, in grams (g).

c—The concentration of sodium hydroxide standard titration solution, in moles per liter (mol/L).

15.999—Molar mass of oxygen, in grams per mole (g/mol).

#### 2.5.4. Mechanical Properties Test

According to GB/T2567-2021 [17], the tensile strength was measured using a universal testing machine (Instron 5967, Instron Corporation, Norwood, MA, USA) at a test speed of 2 mm/min. The sample dimensions were (200 ± 1) mm × (10.0 ± 0.2) mm × (4.0 ± 0.2) mm. During the compression strength testing, the test speed was set at 2 mm/min, and the sample dimensions followed the Type I sample size of (25 ± 0.5) mm × (10 ± 0.2) mm × (10 ± 0.2) mm. The impact strength was measured with a digital display pendulum impact testing machine (XJJ-50, Donglai Instrument Testing Co., Ltd., Chengde City, China), Type II (without notch) sample of dimensions (80 ± 1) mm × (10.0 ± 0.2) mm × (4.0 ± 0.2) mm. The fracture toughness (K_IC_) of the test sample was evaluated according to CB/T41932-2022 [18],with dimensions of 40 × 8 × 4 mm^3^ and a crack length ranging from 3.6 to 4.4 mm. All reported test data represent the average of five samples.

#### 2.5.5. Morphological Analysis

The fracture surface morphology of the impact samples was observed using a field emission scanning electron microscope (Zeiss Ultra Plus, Carl Zeiss, White Plains, NY, USA) at an accelerating voltage of 5 kV. Prior to observation, the fracture surfaces were sputter-coated with gold.

Using an atomic force microscope (AFM: Bruker Dimension ICON, Bruker Corporation, Billerica, MA, USA), the samples were tested in PeakForce Tapping Mode to obtain their morphology and modulus.

#### 2.5.6. Dynamic Thermomechanical Analysis Method

Dynamic mechanical analysis: Using a dynamic mechanical analyzer (DMA 8000, PerkinElmer, Waltham, MA, USA) in bending mode, the sample was heated from 30 °C to 280 °C at a heating rate of 5 °C/min with a frequency of 1 Hz. Sample dimensions: 35 mm × 10 mm × 2 mm.

#### 2.5.7. Thermogravimetric Analysis Method

STA449F3 Jupiter (NETZSCH-Gerätebau GmbH, Selb, Germany) was used for the thermogravimetric analysis with a heating rate of 10 °C/min. The cured resin samples were heated from room temperature to 1000 °C under both air and nitrogen atmospheres.

## 3. Results and Discussion

### 3.1. Fourier Transform Infrared Analysis

Figure 2 shows the FTIR spectra of Z-6018, AG80 epoxy resin, and AG/P. In the FTIR spectrum of Z-6018, the characteristic peaks at 3340 cm^−1^, 1432 cm^−1^, 1130 cm^−1^, and 696 cm^−1^ correspond to the hydroxyl group, benzene ring, Si−O−Si bond, and propyl group, respectively. In the FTIR spectrum of AG80, the peaks at 1614 cm^−1^, 1565 cm^−1^, and 1517 cm^−1^ correspond to benzene ring vibrations. The peaks at 1335 cm^−1^ and 1190 cm^−1^ are attributed to aromatic tertiary amine (Ar–N) and aliphatic tertiary amine (H_2_C−N), respectively, while peaks at 908 cm^−1^ and 830 cm^−1^ indicate epoxy functional groups. Compared to AG80, the broad peak at 1170–1000 cm^−1^ in AG/P originates from the overlap between the characteristic Si–O–C stretching vibration at 1133 cm^−1^ and the Si–O–Si band (1130–1030 cm^−1^) [19,20], with peak intensity slightly increasing with higher Z-6018 content. The characteristic peaks at 908 cm^−1^ and 830 cm^−1^ remain consistent with AG80, though with slightly reduced intensity, indicating a reaction between epoxy groups and Si–OH. Additionally, a new peak that appears at 696 cm^−1^ corresponds to the propyl group. These spectral changes collectively confirm the formation of the AG/P copolymer.

### 3.2. NMR Spectroscopy Analysis

The ^1^H NMR spectrum provides evidence for the chemical reaction between Z-6018 and AG80. In the ^1^H NMR (CDCl_3_) spectrum of AG80 shown in Figure 3, prominent chemical shift peaks include 7.08–6.75 ppm (aromatic protons, dd), 3.81 ppm (Ar–CH_2_–Ar, s), 3.73–3.39 ppm (–CH_2_–epoxy groups, m), 3.19–3.17 ppm (–CH–, epoxy groups, t), and 2.82–2.59 ppm (–CH_2_, epoxy groups, dq). Compared to AG80, the ^1^H NMR spectrum of AG/P shows an overlapping peak at 3.7 ppm for the –CH(OH)– characteristic peak, and the appearance of a peak at 2.06 ppm (CH_2_–O–Si, s) is attributed to the reaction between AG80 and Z-6018 [11,12]. However, the 2.06 ppm characteristic peak is not observed in the AG80 and Z-6018 blend, further confirming the formation of the AG/P copolymer.

As shown in Figure 4, the ^29^Si NMR spectrum of Z-6018 exhibits two characteristic peaks: a Si–O–Si characteristic peak at –111 ppm and Si-OH characteristic peaks at –69 ppm and –89 ppm [11,21]. The ^29^Si NMR analysis of AG/P revealed a significantly reduced intensity of the Si–O–Si resonance peak at –116 ppm compared to Z-6018, while the disappearance of characteristic peaks for Si–OH was observed (at approximately –69 ppm and –89 ppm). This indicates that the terminal hydroxyl groups of Z-6018 reacted with the epoxy groups of AG80. These results are consistent with the substantial decrease in hydroxyl peak intensity observed in the FTIR spectrum of AG/P.

### 3.3. Epoxy Value Analysis

The epoxy value refers to the quantity of epoxy groups in 100 g of epoxy resin. The theoretical epoxy value can be calculated by multiplying the epoxy value of unmodified resin by the epoxy resin content in the modified resin. As shown in Figure 5, the experimentally measured epoxy value of AG/P is lower than the theoretical value, confirming that Si–OH groups in Z-6018 reacted with epoxy groups in AG80, consuming a portion of the epoxy functionalities. According to the experimental results, with the increase in the addition amount of Z-6018, the consumption of epoxy groups also increases, and the epoxy value of AG/P30 decreases by 14.1% compared with that before the reaction. This is attributed to the reaction of Si–OH with more epoxy groups, indicating that the degree of copolymerization between Z-6018 and AG80 is increasing. Epoxy value titration provides supporting evidence for the FTIR and NMR results, further confirming the copolymerization between Z-6018 and AG80. Due to limited reaction progress, the system exists in a copolymer/blend hybrid state.

FTIR spectroscopy and NMR spectroscopy provide molecular-level evidence for the copolymerization between Z-6018 and AG80, characterized by new absorption peaks confirming the reaction between Si–OH and epoxy groups. The decreasing trend in epoxy values macroscopically validates the reaction extent. These mutually corroborating results fully confirm the synthetic route depicted in Figure 1.

Variations in the Z-6018 content lead to changes in the phase structure and macroscopic properties of AG80 epoxy resin. Consequently, we will systematically investigate the effects of different Z-6018 contents on the mechanical properties, phase structure, and thermal stability of AG80. Simultaneously, by introducing P/E30 epoxy compatibilizer at varying concentrations, we will analyze its regulation effect on phase structure.

### 3.4. Mechanical Properties and Morphology Analysis

#### 3.4.1. Mechanical Properties and Morphology Analysis of AG/P

As shown in Figure 6a, the tensile and compressive strengths of AG80 are 26.24 MPa and 141.91 MPa, respectively. When the content of Z-6018 is at 10 wt.% and 20 wt.%, the tensile strength remains relatively stable before and after modification. However, at 30 wt.% content, the tensile strength increases to 30.02 MPa, showing a 14.37% improvement compared to the unmodified resin. In contrast to the change in tensile strength, the compressive strength of the modified resin decreases with increasing Z-6018 content, primarily due to the decrease in crosslink density after curing, a conclusion supported by subsequent DMA testing. It is noteworthy that the tensile strength does not exhibit the same trend as the compressive strength. This is attributed to enhanced molecular chain mobility when cross-linking density is moderately reduced. Under tensile stress, flexible chain segments delay crack propagation through slippage, orientation alignment, and stress dispersion, thereby increasing tensile strength. Conversely, under compressive loads, materials rely on chain rigidity and cross-linking points to resist deformation. Reduced cross-linking density diminishes buckling resistance, making chains prone to shear slip or volumetric collapse under pressure. As shown in Figure 6b, the impact strength of AG80 is 2.19 kJ/m^2^. With increasing Z-6018 content, the impact strength of the modified resin progressively improved. When the content reaches 30 wt.%, there is a 121.92% enhancement compared to the unmodified resin. To further confirm the enhanced toughness of the modified resin, Figure 6b shows that fracture toughness (K_IC_) progressively increases with higher Z-6018 content. Compared to the unmodified resin, the K_IC_ of AG/P30 increased by 65.05%, which is consistent with the trend of impact strength change.

To investigate the influence of microscopic morphology on material toughness, morphological analysis of impact fracture surfaces was conducted using SEM. In Figure 7a, the cured fracture surface of AG80 exhibits a smooth microstructure, characteristic of typical brittle thermosetting resins. In contrast, the modified resin fracture surface shows a distinct “sea island structure” and surface roughening, confirming a transition from brittle to tough fracture behavior. Based on our team’s prior research [11], the silicon-rich spherical phase consists of a Z-6018-dominated copolymer, while the epoxy resin matrix comprises an epoxy-based copolymer. Figure 7b–d demonstrate that with increasing Z-6018 content, the number of silicon-rich spherical phases gradually rises, and the “sea island structure” becomes more pronounced. This morphological feature is responsible for the enhanced material toughness [7].

Further research reveals that the quantity, particle size, and distribution of silicon-rich spherical phases are critical factors contributing to variations in the mechanical properties of modified resins. As shown in Figure 8a,b, increasing Z-6018 content promotes the growth of silicon-rich spherical phases, with both average particle size and size distribution breadth rising. At a content of 30 wt.%, the average particle size reaches 3.10 μm, distributed between 1.2–4.4 μm. This is due to the increase in the content of polysiloxane, which leads to a higher probability of aggregation, thereby promoting the growth of silicon-rich spherical phases. Our team modified bisphenol A epoxy resin with Z-6018, achieving a 34.3% increase in tensile strength [12]. Although AG/P30 outperformed AG80 by 14.37%, it still fell short of the expected enhancement. Prior studies revealed that excessive Z-6018 induces internal stress concentration, triggering rapid crack propagation along phase interfaces and paradoxically weakening reinforcement [11]. This indicates that the compatibility between silicon-rich spherical phases and the matrix in AG/P30 requires further optimization.

#### 3.4.2. Mechanical Properties and Morphology Analysis of AG/P30-S

The P/E30 molecular structure contains silanol and epoxy groups, serving as a bridge to connect the polysiloxane and epoxy resin phases, thereby enhancing their compatibility. The phase structure was regulated by introducing P/E30 to investigate its effect on the mechanical properties of materials. As shown in Figure 9, the mechanical test results of AG/P30-S demonstrate that P/E30 introduction significantly enhances toughening and strengthening effects: at a content of 20 wt.%, the tensile strength reached 39.59 MPa, and the impact strength reached 12.15 kJ/m^2^, representing increases of 50.89% and 454.79% compared to AG80, respectively. Notably, P/E30 enhances tensile strength while maintaining stable compressive strength, likely due to compatibility improvements that regulate the cured cross-linked network structure.

To further elucidate the compatibility enhancement by P/E30, morphological analysis of the impact fracture surface for AG/P30-S was performed using SEM. In Figure 10a–c, the fracture morphology of AG/P30-S20 differs from that of AG/P30-S10 and AG/P30-S30: in AG/P30-S20, the silicon-rich spherical phase exhibits an indistinct interface with the matrix and fractures without cavitation. Conversely, in AG/P30-S10 and AG/P30-S30, the spherical phases are pulled out from the matrix, forming cavities. This confirms stronger interfacial interactions in AG/P30-S20 [22,23]. This indicates that the content of 20 wt.% P/E30 can effectively promote connectivity between polysiloxane and epoxy resin phases, thereby reducing interfacial issues induced by phase separation during curing. However, when the P/E30 content exceeds a threshold, the excessive increase in silicon-rich spherical phases adversely affects material performance. Compared to AG/P30-S20, the quantity of silicon-rich spherical phases in AG/P30-S30 is increased by 150%, an excess of which can induce the formation of internal defects and weaken interfacial interactions [24]. Under external stress, silicon-rich spherical phases readily debond from the matrix and are pulled out to form cavities. The energy dissipated by interfacial debonding cavitation is low, and its high-density distribution degrades mechanical properties [25]. This explains the lower mechanical properties of AG/P30-S30 compared to AG/P30-S20. The tensile strength and toughness of materials are closely related to the interfacial bonding state. Stronger interfacial interactions facilitate stress transfer, thereby enhancing both tensile strength and toughness [13]. Figure 11 demonstrates that weak interfacial bonding induces debonding between silicon-rich spherical phases and the matrix. This not only restricts the elastic deformation capacity of the spherical phases but also promotes preferential crack propagation along phase interfaces. Conversely, enhanced interfacial bonding forces the crack tip to penetrate the silicon-rich spherical phases, thereby inducing crack branching and termination. AG/P30-S10 and AG/P30-S30 exhibit limited microcracks on fracture surfaces due to weak interfacial interactions, while AG/P30-S20 shows branched and terminated microcracks. During the propagation process, cracks deviate from their original path and extend in different directions, increasing the fracture area [26]. As cracks extend into silicon-rich spherical phases with higher toughness, these phases effectively alleviate stress concentration at the crack tip. This promotes crack tip blunting and termination. The process substantially dissipates energy, thereby enhancing both the toughness and strength of the material.

Figure 12a–d demonstrate that after introducing P/E30, the average particle size of silicon-rich spherical phases gradually decreased from 3.10 μm to 1.21 μm, while the size distribution narrowed from 1.2–4.4 μm to 0.7–1.7 μm. Acting as a bridge between polysiloxane and epoxy phases, P/E30 promotes uniform dispersion of silicon-rich spherical phases in the AG80 resin matrix, leading to a reduction in particle size and distribution width, thereby enhancing system compatibility and enabling the regulation of phase structure.

The analysis above indicates that merely increasing Z-6018 content results in larger silicon-rich spherical phases and weaker phase boundaries, with limited enhancement in mechanical properties. To address this, the compatibilizer P/E30 is introduced, which functions by (1) suppressing polysiloxane aggregation to promote uniform dispersion; (2) forming silicon-rich spherical phases of suitable size and quantity; (3) strengthening interfacial interactions to reduce the debonding of silicon-rich spherical phases. Therefore, optimizing the compatibilizer content can achieve dual regulation of silicon-rich spherical phase size and phase boundaries, leading to the toughening and strengthening of the epoxy resin.

#### 3.4.3. AFM Phase Structure Analysis

Further analysis of the phase structure was conducted using AFM. Figure 13a demonstrates the AFM morphology of cured AG80, AG/P30, and AG/P30-S20 samples. AG/P30 exhibits the significant size variations and non-uniform distribution of silicon-rich spherical phases. Upon the introduction of 20 wt.% P/E30, the particle size distribution of silicon-rich spherical phases improves, showing good dispersion, consistent with the SEM results. As shown in Figure 13b,c, AG/P30-S20 exhibits a smaller height difference between silicon-rich spherical phases and the matrix than AG/P30, indicating enhanced interfacial bonding. These results confirm that introducing 20 wt.% P/E30 effectively improves interfacial compatibility.

Derjaguin–Muller–Toporov modulus (DMT modulus) analysis characterized the modulus features of silicon-rich spherical phases. Figure 14a displays the DMT modulus maps of cured AG80, AG/P30, and AG/P30-S20, with modulus values visualized by color contrast: bright regions correspond to high-modulus hard phases, while dark regions indicate low-modulus soft phases. AG80 exhibits a uniform elastic modulus distribution across its surface. The modulus of AG/P30 is lower than that of AG80, with a distinct interface between the hard and soft phases. The content of 20 wt.% of P/E30 results in a reduced difference in modulus between the two phases, leading to a blurred phase interface, consistent with the observed trend of phase fusion under SEM. Figure 14b,c demonstrate that the AG/P30 silicon-rich spherical phase region exhibits minor modulus fluctuations, whereas the AG/P30-S20 exhibits significant modulus variations. This phenomenon arises from the introduction of P/E30, leading to the formation of a bicontinuous phase structure within the silicon-rich spherical phases of AG/P30-S20. As illustrated in Figure 15, within the bicontinuous phase structure, the high modulus region bears loads by the epoxy resin continuous phase, while the low modulus region dissipates stress by elastic deformation of the epoxy–polysiloxane copolymer/blend dispersed phase. This structural feature enhances the overall mechanical properties of the material.

### 3.5. Thermal Performance Analysis

#### 3.5.1. Dynamic Mechanical Analysis

The dynamic mechanical properties of AG/P and AG/P30-S20 were investigated by DMA, with the results presented in Figure 16 and relevant data compiled in Table 2. Figure 16a shows the tanδ curves of AG/P and AG/P30-S20, where the peak temperature corresponds to the glass transition temperature (Tg) of the materials. Typically, the addition of silicone reduces the Tg of cured thermosetting resins [27]. As the Z-6018 content increases from 0 to 30 wt.%, the Tg of the modified resin decreases from 239 °C to 204 °C. This change is mainly due to the introduction of flexible Si–O–Si chain segments and the reduction in crosslink density. According to rubber elasticity theory [26], the crosslink density (ρ) of cured resin is calculated by ρ = E’/(3RT), where E’ is the storage modulus at Tg+50K, R is the gas constant (8.314 J·mol^−1^·K^−1^), and T is the absolute temperature at Tg+50K. As shown in Table 2, the crosslink density of cured resins progressively decreases with increasing Z-6018 content. This is mainly attributed to (1) the decrease in epoxy group content due to the opening reaction between the epoxy resin and Si–OH; (2) the formation of a Si–O–Si network, reducing the crosslinking density; (3) the excessive size of silicon-rich spherical phases can disrupt the resin’s crosslinking structure, leading to a decrease in crosslinking density [28]. Notably, compared to AG/P30, AG/P30-S20 exhibits a decrease in the size of silicon-rich spherical phases, a reduction in tanδ peak, an increase in Tg, and an increase in crosslinking density. This change is attributed to the decrease in the size of silicon-rich spherical phases, which enhances the dispersion of polysiloxane in the cross-linked structure of AG80 epoxy resin, thereby increasing the system’s crosslinking density and suppressing the lowering effect of flexible Si–O–Si chain segments on Tg [29].

As shown in Figure 16c, the modified resin exhibits a higher loss modulus (E”) than AG80 within 30–200 °C. This is attributed to the increased number of Si–O–Si chain segments in the cross-linked network, enhancing the mobility of Si–O–Si and Si–O–C flexible chain segments. This helps to convert fracture energy into heat energy, thereby improving the material’s resistance to damage under room temperature vibration and impact loads, enhancing the toughness of the modified resin [26,30].

#### 3.5.2. Thermogravimetric Analysis

Figure 17 compares the TGA and DTG curves of different cured resins under air and nitrogen atmospheres, while Table 3 summarizes the initial thermal degradation temperature, maximum thermal degradation temperature, and char yield at 1000 °C for each system in both atmospheres. According to Table 3, the introduction of Z-6018 results in the T5% of AG/P resin being lower than that of AG80 epoxy resin. This is attributed to two main factors: (1) terminal hydroxyl groups on Z-6018 initiate chain scission preferentially upon heating. This low activation energy process results in earlier thermal degradation of AG/P resin at lower temperatures [31,32]; (2) reduced AG80 epoxy content decreases crosslink density, weakening intermolecular forces and lowering the activation energy required for thermal degradation, thereby accelerating weight loss [8,11,33]. Char yield data indicate that after grafting with Z-6018, the char yields of AG/P30 at 1000 °C reached 9.98% in air and 23.17% in nitrogen, representing increases of 255.16% and 172.59% compared to AG80 resin, respectively. This enhancement is attributed to the low surface energy of polysiloxane, which migrates to the resin surface during heating, forming a silicon-rich char layer that insulates heat and slows down resin thermal degradation [34,35]. Additionally, Figure 17c,d demonstrate that the maximum decomposition rates in DTG curves decrease with increasing Z-6018 content, further indicating that the introduction of Z-6018 slows down the resin’s thermal degradation rate, thereby enhancing thermal stability. Notably, the addition of P/E30 increased the initial and maximum thermal degradation temperatures of AG/P30, with minimal impact on the char yield. This enhancement can be attributed to the improved compatibility between polysiloxane and AG80 epoxy resin, resulting in higher curing crosslink density and strengthened intermolecular interactions, as confirmed by DMA analysis.

## 4. Conclusions

This study synthesized AG/P resin using Z-6018 and AG80 epoxy resin, investigated the effect of P/E30 content on the phase structure of the modified resin, and elucidated the relationship between phase structure and macroscopic properties. The research findings indicate that the presence of large and unevenly distributed silicon-rich spherical phases in modified resins has a limited impact on improving mechanical properties. To enhance toughness and strength, the regulation of resin phase structure was achieved by introducing P/E30. At 20 wt.% P/E30 addition, the tensile strength, impact strength, and fracture toughness increased by 50.89%, 454.79%, and 152.43%, respectively, compared to AG80. P/E30 enables the synergistic enhancement of strength and toughness in AG/P30-S20 by modulating the phase structure. This is attributed to (1) effective control of the size and distribution uniformity of silicon-rich spherical phases; (2) strengthened interfacial interactions that promote stress transfer, thereby improving tensile strength; and (3) the formation of a bicontinuous phase structure within the silicon-rich spherical phases. In this structure, the high modulus region bears load by the epoxy resin continuous phase, and the low modulus region dissipates stress by elastic deformation of the epoxy–polysiloxane copolymer/blend dispersed phase, achieving increased toughness and strength. The toughening and strengthening of modified resins depend not only on phase structure but also on crosslinking density. The introduction of flexible Si-O-Si chain segments reduces crosslinking density, lowering the Tg while enhancing molecular chain mobility, thereby improving the material’s toughness. 20 wt% P/E30 compatibilizer increased the crosslinking density of the modified resin, enhancing tensile strength while retaining the toughening effect of Si–O–Si chain segments. Thermogravimetric analysis revealed that the char yield of the modified resin at 1000 °C increased by 255.16% in air and 172.59% in nitrogen atmosphere compared to AG80 resin. This is attributed to the high-temperature migration of polysiloxane to the surface, forming a silicon-rich char layer that insulates heat and slows down resin thermal degradation. By elucidating the relationship between phase structure and macroscopic properties, the proposed phase structure regulation strategy achieves a breakthrough in balancing strength and toughness, which has long been a critical challenge. This approach can be extended to other epoxy/polysiloxane systems, providing a viable strategy for developing resin matrices with balanced high strength, toughness, and thermal stability.

## Figures and Tables

**Figure 1 polymers-17-02099-f001:**
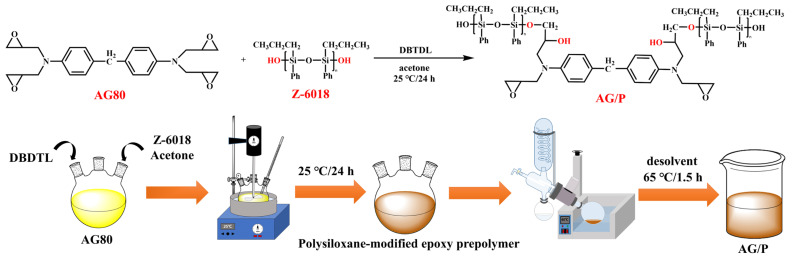
Flowchart of AG/P synthesis.

**Figure 2 polymers-17-02099-f002:**
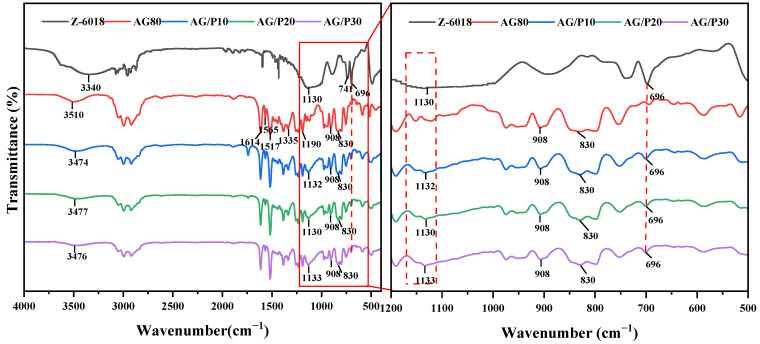
FTIR spectra of Z-6018, AG80, and AG/P.

**Figure 3 polymers-17-02099-f003:**
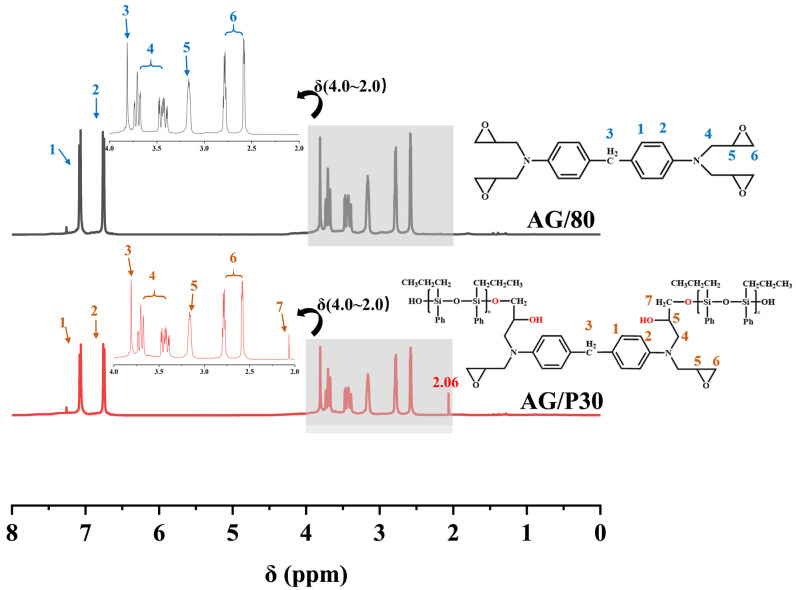
^1^H NMR spectra of Z-6018 and AG/P.

**Figure 4 polymers-17-02099-f004:**
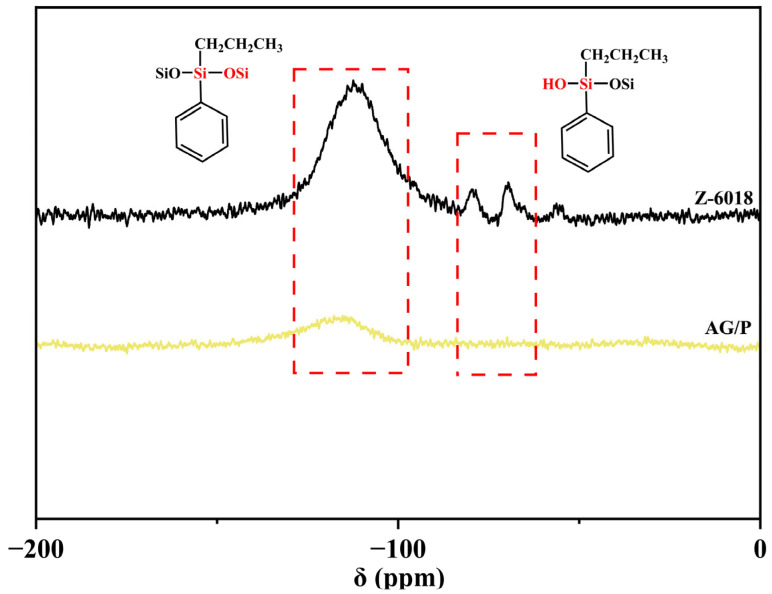
^29^Si NMR spectra of Z-6018 and AG/P.

**Figure 5 polymers-17-02099-f005:**
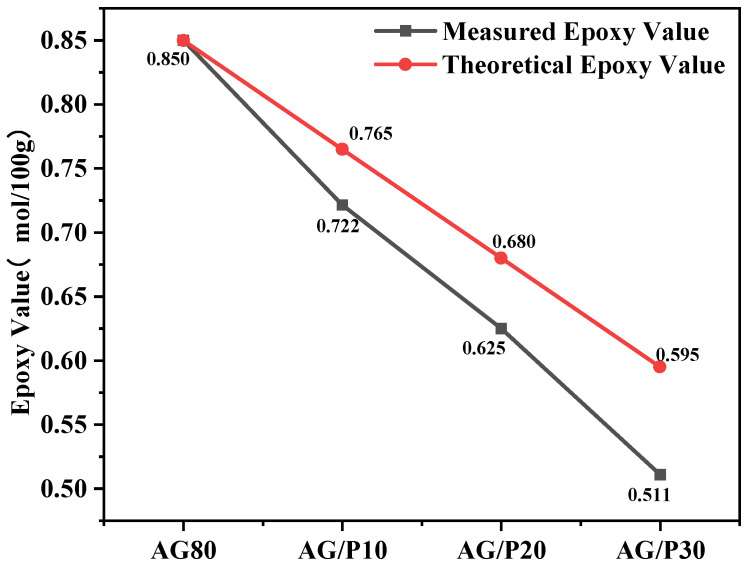
Theoretical and measured epoxy values.

**Figure 6 polymers-17-02099-f006:**
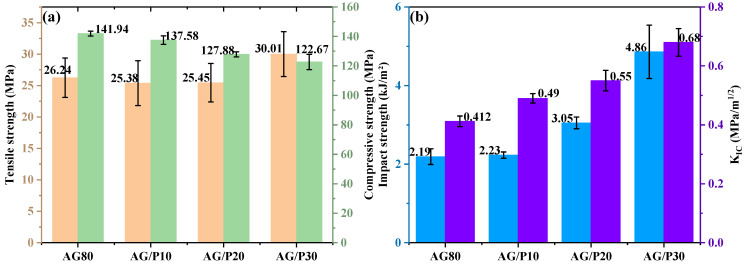
Mechanical properties of AG/P: (**a**) tensile and compressive strength; (**b**) impact strength and K_IC_.

**Figure 7 polymers-17-02099-f007:**
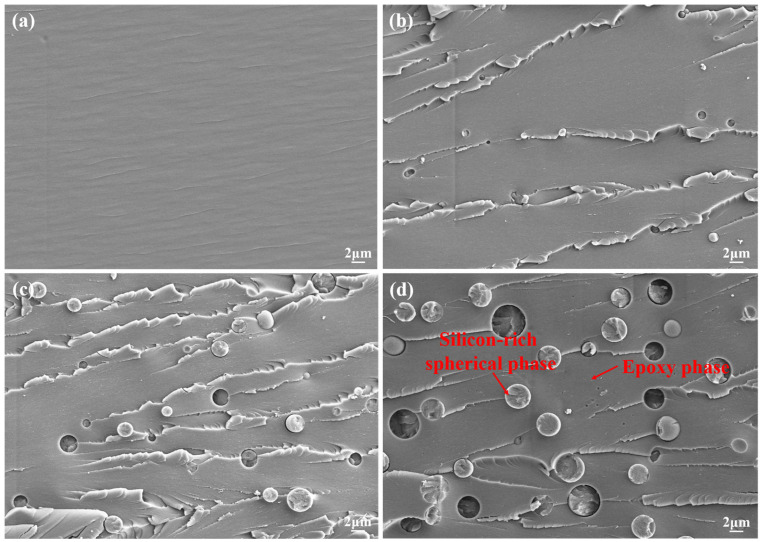
SEM images of impact fracture surfaces (2000×): (**a**) AG80; (**b**) AG/P10; (**c**) AG/P20; (**d**) AG/P30.

**Figure 8 polymers-17-02099-f008:**
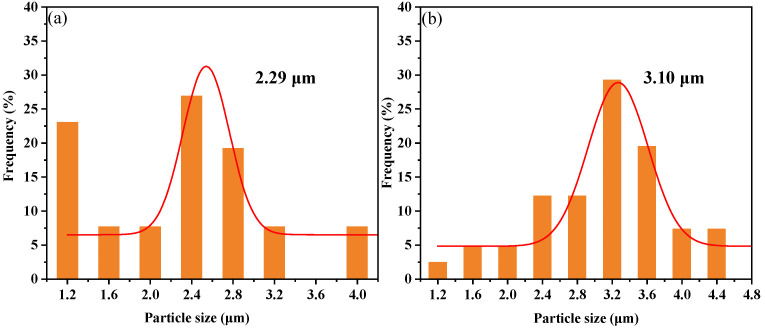
Particle size distribution of silicon-rich spherical phase: (**a**) AG/P20; (**b**) AG/P30.

**Figure 9 polymers-17-02099-f009:**
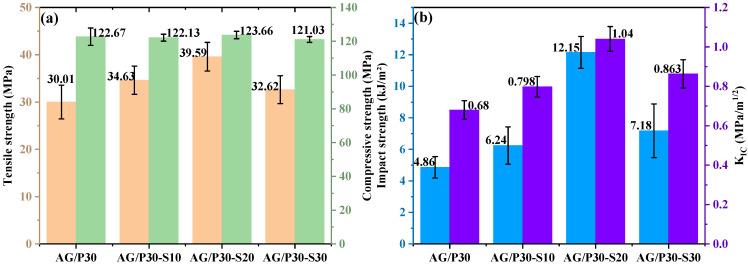
Mechanical properties of AG/P30-S: (**a**) tensile and compressive strength; (**b**) impact strength and K_IC_.

**Figure 10 polymers-17-02099-f010:**
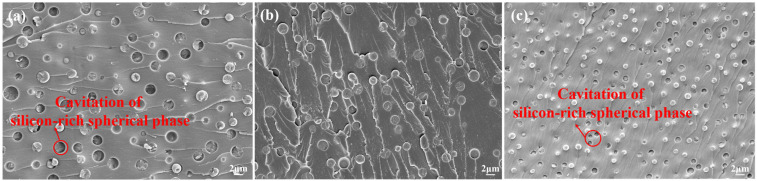
SEM images of impact fracture surfaces (2000×): (**a**) AG/P30-S10; (**b**) AG/P30-S20; (**c**) AG/P30-S30.

**Figure 11 polymers-17-02099-f011:**
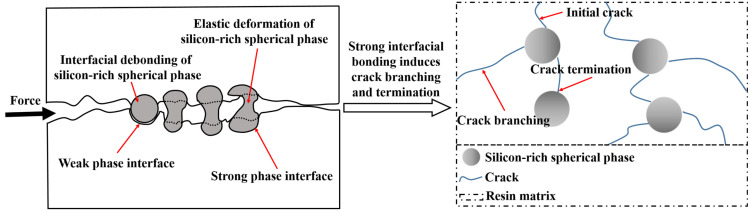
Strong interfacial bonding induces crack branching and termination.

**Figure 12 polymers-17-02099-f012:**
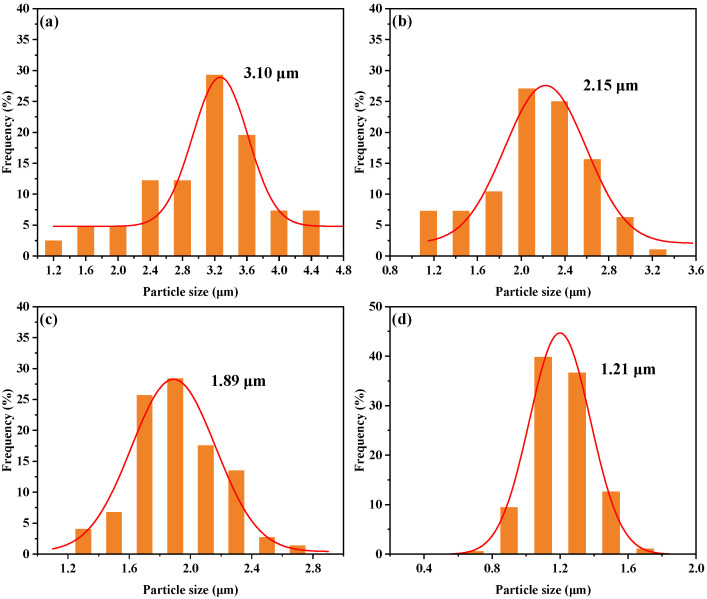
Particle size distribution of silicon-rich spherical phase: (**a**) AG/P30; (**b**) AG/P30-S10; (**c**) AG/P30-S20; (**d**) AG/P30-S30.

**Figure 13 polymers-17-02099-f013:**
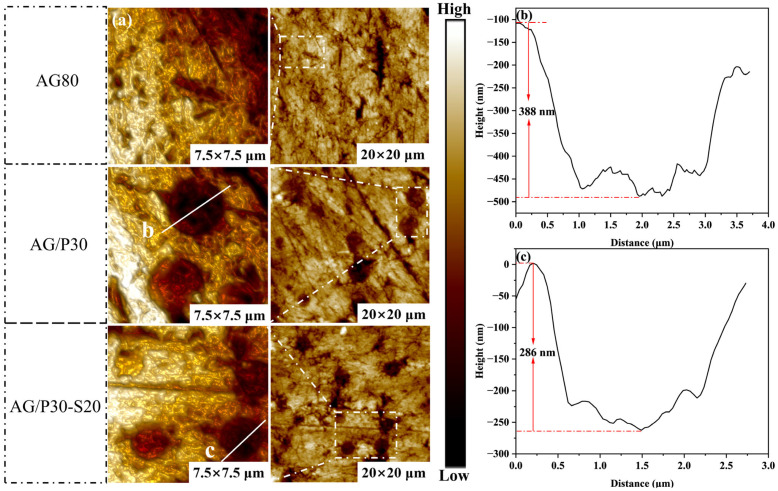
AFM morphology characterization results: (**a**) morphology images of AG80, AG/P30, and AG/P30-S20; (**b**) height profile at the interface between silicon-rich spherical phase and matrix in AG/P30; (**c**) height profile at the interface between silicon-rich spherical phase and matrix in AG/P30-S20.

**Figure 14 polymers-17-02099-f014:**
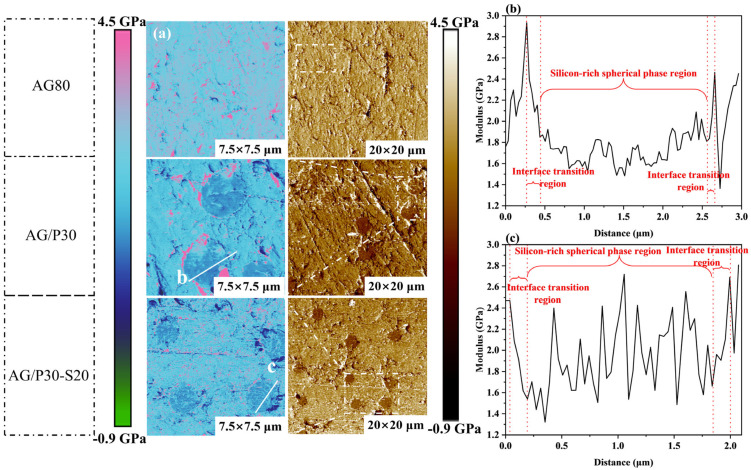
AFM modulus characterization results: (**a**) DMT modulus maps of AG80, AG/P30, and AG/P30-S20; (**b**) modulus profile across the silicon-rich spherical phase in AG/P30; (**c**) modulus profile across the silicon-rich spherical phase in AG/P30-S20.

**Figure 15 polymers-17-02099-f015:**
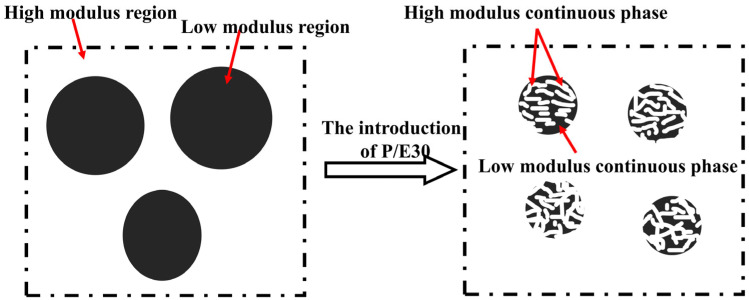
Formation of bicontinuous phase structure within the silicon-rich spherical phase.

**Figure 16 polymers-17-02099-f016:**
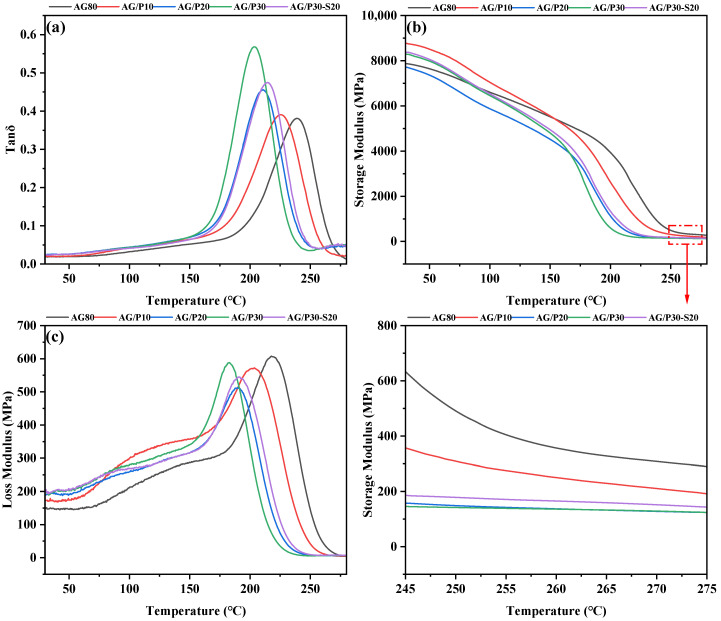
DMA curves of cured AG/P and AG/P30-S20 resins: (**a**) loss factor; (**b**) storage modulus; (**c**) loss modulus.

**Figure 17 polymers-17-02099-f017:**
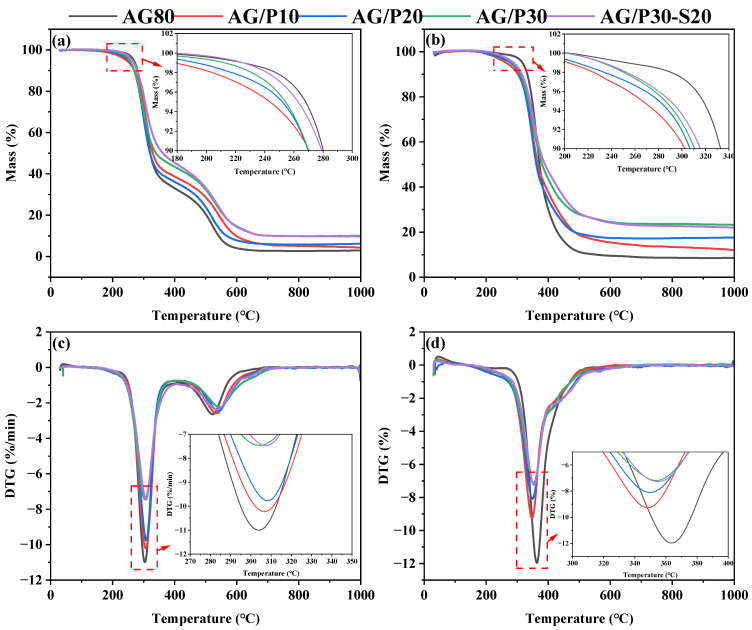
Thermal analysis curves of cured AG/P and AG/P30-S20 resins under air and nitrogen atmospheres: (**a**) TG results under air atmosphere; (**b**) TG results under nitrogen atmosphere; (**c**) DTG results under air atmosphere; (**d**) DTG results under nitrogen atmosphere.

**Table 1 polymers-17-02099-t001:** Resin curing formulation.

Sample	AG80/g	Z-6018/g	DMP-30/g	4-MHHPA/g	P/E30/g
AG80	100	0	0.5	142.97	0
AG/P10	90	10	0.5	121.61	0
AG/P20	80	20	0.5	108.49	0
AG/P30	70	30	0.5	85.95	0
AG/P30-S10	63	27	0.5	114.86	10
AG/P30-S20	56	24	0.5	108.12	20
AG/P30-S30	49	21	0.5	101.37	30

**Table 2 polymers-17-02099-t002:** DMA data of cured resin.

Sample	Phase Size/μm	Tg/°C	tanδ	E’/MPa	ρ × 10^−3^/(mol/m^3^)
AG80	--	239	0.36	258	18.4
AG/P10	--	225	0.36	189	13.8
AG/P20	2.29	210	0.41	136	10.2
AG/P30	3.10	204	0.53	130	9.8
AG/P30-S20	1.89	214	0.43	159	11.9

**Table 3 polymers-17-02099-t003:** Thermogravimetric data of cured resins.

Sample	Air				N_2_		
	T_5%_/°C	T_1max%_/°C	T_2max%_/°C	R_1000 °C_/%	T_5%_/°C	T_max%_/°C	R_1000 °C_/%
AG80	268	303	522	2.81	317	363	8.50
AG/P10	264	306	532	4.31	266	348	12.07
AG/P20	261	308	540	6.07	280	349	17.54
AG/P30	259	304	545	9.98	285	354	23.17
AG/P30-S20	262	307	535	9.53	289	355	21.93

Note: T_5%_ represents the initial thermal degradation temperature; T_1max_ and T_2max_ denote the maximum thermal degradation temperatures for the first and second stages, respectively; R_1000 °C_ indicates the residual char yield at 1000 °C.

## Data Availability

Data is contained within the article.

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
