# Peer review of "Effects of Phase Structure Regulation on Properties of Hydroxyl-Terminated Polyphenylpropylsiloxane-Modified Epoxy Resin"

_polymers, 2025, doi:10.3390/polym17152099_

Round 1
Reviewer 1 Report
Comments and Suggestions for Authors
The authors presented a very interesting work devoted to studying the effect of the phase structure of a new modified epoxy resin on its strength and thermal properties. The presented study is very well thought out and well implemented, the results are relevant for modern polymer materials science.
There are a number of small comments on the work:
- For which of the synthesized AG/P is the FTIR spectrum shown in Fig. 3? Were all the compositions studied? Was the effect of the Z-6018 concentration visible? How was the study conducted? Transmission or ATR?
- Also, the authors should be more careful in interpreting the absorption peaks in the IR spectra: after all, the peak at 3340 is not Si-OH, but simply OH. The same is true for Si-C6H5.
- It is surprising that the absorption band of hydroxyl groups in the spectrum for AG/P is smaller than that of the resin, although the number of OH groups should clearly have increased. Where did the hydroxyl groups in the original resin come from?
- Where do the bands at 3700-3500 come from in the IR spectrum of Z-6018? Amines?
- Fig. 2 and 3 could have been made larger so that the signatures and diagrams could be seen. Also, in Fig. 2, similar to Fig. 3, it would have been possible to make an insert of a larger size with the spectra superimposed on each other to compare the most indicative part of the spectrum (for example, the one where the decomposition of some epoxy groups and the formation of additional bonds with Si are visible). This is poorly visible in the presented spectrum.
- The description of Fig. 4 (lines 246-253) raises questions. The authors say (and Fig. 4 shows) that the peak at -111 ppm associated with Si-O-Si is significantly reduced and shifted to -116. But the Si-O-Si bonds remained unchanged in AG/P according to the general reaction scheme in Fig. 1 (Z-6018 is simply incorporated into the exposed epoxy group of the resin). The authors should explain in more detail what they mean.
- The size of the numbers and text in most of the figures is too small to be understood (especially in Fig. 6, 9, 11, 14, 16, 17).
- According to Fig. 7, AG/P has a matrix-inclusion phase structure. It is indicated that the matrix is an epoxy resin. In fact, it was written above that the system has two components - the newly synthesized AG/P resin and the remains of Z-6018 that is not incorporated into the resin. The authors should explain in more detail what exactly is the matrix and what is the dispersed phase, since they previously said that the synthesis occurred with the production of a new material.
- It remains unclear what exactly the compatibilizer was, what its chemical structure was, and how exactly it interacted with the resin and Z-6018.
Reviewer 2 Report
Comments and Suggestions for Authors
This manuscript presents the material characterization of modified AG80 epoxy resins to identify the effect of hydroxy-terminated polyphenylpropylsiloxane (Z-6018) and a self-synthesized epoxy compatibilizer (P/E30). Overall, the results from this investigation are quite interesting, and this manuscript is easy to follow and understand. However, several issues in the manuscript require careful attention before it can be reconsidered for publication. Please revise the manuscript based on the comments as follows:
- The abstract is somewhat overlong. Please shorten the abstract, particularly the result part. Although the results section is well-written, especially when it includes quantitative findings, I suggest that the authors please shorten the explanation and discussion.
- For the keywords, it is recommended not to use the same words that appear in the main title.
- The novelty of this work is unclear. As mentioned in the introduction, various polysiloxane-modified resins have shown enhanced toughness and thermal stability. So, what is the novelty of this work? Please highlight the novelty in the last paragraph of the introduction.
- In section 2.1, please provide the states and countries for all the suppliers of the materials.
- Please mention Figure 1 in the text.
- In section 2.5.6, please provide the temperature range for DMA.
- On page 9; lines 289 – 290, the authors ascribe the reduction in compressive strength to the decrease in crosslink density after curing. However, it is not understandable why the tensile strength did not show the same trend.
- On page 10, lines 319–321, indeed, aggregation may lead to higher stress concentrations, causing rapid crack propagation. This scenario was supposed to reduce the mechanical properties of the materials. However, the highest tensile strength, impact strength, and fracture toughness were all observed in AG/P30. The authors must provide a plausible explanation for this trend.
- Please improve the resolution of Figure 13.
- On page 13; line 413, please write out the full word of ‘DMT’ for its first instance.
- Please improve the resolution of Figure 14.
- Please revise the conclusion. The conclusion should highlight the major findings from the investigation. Please write the conclusion precisely, highlighting the major findings and contribution of the work. Additionally, the last paragraph should be combined with its previous paragraph, as it consists of only one sentence.

Round 2
Reviewer 2 Report
Comments and Suggestions for Authors
The authors have carefully revised the manuscript based on the reviewer comments. This manuscript can now be accepted for publication.